# Attitudes towards Slum Tourism in Mumbai, India: Analysis of Positive and Negative Impacts

**António Cardoso** [1,*] **, Amândio da Silva** [1,2] **, Manuel Sousa Pereira** [3] **, Neeta Sinha** [4] **, Jorge Figueiredo** [5] **and Isabel Oliveira** [5]

[1] Department of Business and Communication Sciences (DBCS), University Fernando Pessoa, 4294-004 Porto, Portugal

[2] GOVCOPP, School of Accounting and Administration (ISCA), University of Aveiro, 3810-193 Aveiro, Portugal

[3] Higher School of Business Sciences (ESCE), Polytechnic Institute of Viana do Castelo, 4930-600 Valença, Portugal

[4] School of Liberal Studies (SLS), Pandit Deendayal Energy University, Gandhinagar 382007, India

[5] Lusiada University, 4369-006 Porto, Portugal

\* Correspondence: ajcaro@ufp.edu.pt

**Abstract:** Tourism has grown exponentially in the 21st century and continues to be one of the rapidly growing industries in the world in terms of revenue generation and employment opportunities. It covers not only travel services and boarding-lodging activities but a wide range of independent but related sectors like transport, accommodation, food and beverage, and entertainment, among others. Modern tourism is diversified and includes several odd types of tourism, like slum tourism, dark tourism, and sex tourism. This paper analyzes the case of slum tourism to Dharavi, India's commercial capital and largest city as well as the benefits and disadvantages that such kind of tourism has. It also attempts to understand the opinion of the common people and slum dwellers on slum tourism, while observing if the ten principles of the "Global Code of Ethics for Tourism" (GCET) have been fulfilled in the country. The results show that overall, the principles of GCET are fulfilled but much is still left to be done. On the other side, most of the slum residents accept slum tourism as a reality that brings more benefits than damage to their living environment and are of the opinion that tourism brings prosperity to them and to the country.

**Keywords:** slum tourism; slum dweller; slum tourist; Dharavi; impacts; Global Code of Ethics for Tourism

## 1. Introduction

Air travel proved to be a blessing for global tourism. Tourism became a cash cow for every country in the world, helping boost consumption, leading to the opening of new hotels and related businesses, and providing new jobs to millions of people [1]. Although the positive effects of tourism are many, it has its negative impacts too, including irreparable damages to the environment and societies, inflation, and ethical concerns.

The tourism sector has tried to implement concepts like ethics, corporate social responsibility, and the triple bottom line, but not every country respects these concepts, this is even more evident in the case of odd and uncommon types of tourism, that are gaining popularity, like slum tourism, sex tourism, and dark (thanatourism), to name just a few. As compared to other uncommon types of tourism, slum tourism is the most popular and is gaining popularity around the globe. It is the only type of odd tourism that families can enjoy together without any kind of embarrassment (as in the case of sex tourism) or fear (in the case of dark tourism).

This article analyzes the benefits and disadvantages of slum tourism in Dharavi, the largest slum in Asia, located in the city of Mumbai, the commercial capital of India, and

where we attempt to measure the implementation of the ten principles of GCET. In addition to being the largest slum in India, Mumbai's Dharavi is the only city in the country that has organized slum tours. Every Indian city has slum dwellers, but none of them has ever witnessed slum tourism activities, thus our study is limited to this area of India.

## 2. Literature Review

### 2.1. The Beginning

As per the United Nations, a slum dweller is one that lives in a tiny, poorly built house that does not have adequate access to clean water and sanitation, in an overcrowded space, without holding a legal title to the property, in a so-called slum [2].

Although slum tourism has existed since the Victorian era, around London's poor East End, it only became an organized concept in the twentieth century [3]. It was in the nineties of the last century that slum tourism started as an organized industry in cities such as Cape Town (South Africa), Rio de Janeiro (Brazil), Mumbai and Delhi (India), Mexico City (Mexico), Nairobi (Kenya), Windhoek (Namibia), Manila (Philippines), among other cities [4].

There are several different types of slum tourism, including the regular tours to a garbage dump organized by a multidenominational American expatriate church in Mazatlán, Mexico [5] and heritage tours in Soweto, Johannesburg, South Africa [6], to tours in Mumbai's Dharavi, to see the squalor and lack of hygiene of the real Indian, living his days in utter misery.

Among the several types of tourism, slum tourism, also known as poverty tourism, township tourism, slumming, poorism, or philanthropic tourism [7] is a controversial and fast-growing type of tourism, in which tourists spend some time, from a couple of hours to a few days, visiting, touring, and experiencing the way of living of locals in shantytowns [8]. It is a growing phenomenon in several developing [9], but has been observed even in the United States, such as in Louisiana after Hurricane Katrina [10] and on the homeless tours in Europe [11]. It has evolved from being practiced in a limited number of countries to becoming a truly global phenomenon performed presently on five continents. There has been an increase in the variety of services and ways in which tourists visit the slums [12], with tour operators increasing their options and offering packages.

Most types of unorganized tourism lack ethics and are run in an unorganized way. This gives an opportunity for illegal elements to cheat tourists and even rob them. So, in order to bring some ethicality into the sector, UNWTO [13] proposed the GCET (Global Code of Ethics for Tourism), basic guidelines for governments, the travel industry, communities, and tourists alike in an attempt to maximize the benefits of the sector for one and all and to minimize the negative impact of tourism on the environment, cultural heritage, and societies across countries. The ten principles of GCET are:

1.  Tourism must contribute to mutual understanding and respect between people and society.
2.  Tourism should be a vehicle for individual and collective fulfillment.
3.  Tourism is a factor for sustainable development.
4.  Tourism is a user of the cultural heritage of mankind and a contributor for its enhancement.
5.  Tourism should be a beneficial activity for the host country and its communities.
6.  Stakeholders' have obligations in tourism development.
7.  Tourism has rights.
8.  There should be liberty and freedom of tourists' movements.
9.  Workers and entrepreneurs in the tourism industry have rights.
10. The global code of ethics for tourism should be implemented by every country.

Although most countries have implemented the GCET guidelines to some extent, the sector is composed of large companies and SMEs operating in the unorganized sector, and it is not possible to manage the ethical issues of such a complex supply chain [14], more so in the case of slum tourism in a large, overpopulated country like India.

### 2.2. Dharavi: Asia's Largest Slum

India is a fast-developing country, that has the second highest population in the world, mostly living in villages and small towns. Rapid urbanization and economic vibrancy of large cities that offer diverse employment opportunities and better means of livelihood attract rural migrants to them, these migrants transfer rural poverty to urban areas. This results in the development of slum areas in cities that lack physical infrastructure in terms of planned housing, drinking water supply, drainage, etc., as the poor cannot afford housing in places where there is a rapid increase in land prices [15].

Asia's largest slum area, Dharavi, is in Mumbai, India's commercial capital and largest city. Dharavi is half the size of New York's Central Park, housing about 1 million people mostly living in spaces under 10 m$^2$, making it over six times denser than New York's Manhattan. The government of India is perpetually trying to relocate the residents of Dharavi to other areas, as the land in which it is located is premium real estate, but all efforts have failed [16].

Slum tourism in Dharavi, Mumbai, was started officially in 2006 by the only tour operator in this field, Reality Tours and Travels, to introduce the city's poverty to foreign tourists and has been growing ever since [17]. This growth can be partly attributed to the popularity of the film *Slumdog Millionaire* which told the story of a young boy from Mumbai's slums [18]. However, not much of the income collected from slum tourists percolates to the slum dwellers, and whatever does, is mostly indirectly in the form of temporary jobs, small business revenue from sales of trinkets to tourists, or even begging [19].

### 2.3. The Economics of Slum Tourism

Tourism is the largest industry in the world, in terms of revenue collected as well as the number of people involved [20], and it has far-reaching implications for the economic growth and welfare of a nation, if properly managed [21] or could bring complications and negatively impact the host community, with authentic cultural representations disregarded or commodified, giving the impression of oppression and suffering beyond the reality, especially in the case of niche tourism like slum [20]. In total, 80% of the slum tours are concentrated in just two destinations: the townships of South Africa and the favelas of Brazil [22], but cities like Mumbai are becoming more and more popular over time.

Although the study of Kenyan slum tourism, by Kieti and Magio [7] proved that the benefits of slum tourism were insignificant to make residents support its further development, the negative attitude towards it has not reached a level where the majority of the residents of the region would oppose it, as it is still expected that its future could be bright. One of the reasons why slum tourism does not bring more benefits is because of the limited opportunities that slum dwellers have to interact with slum tourists and the perpetual "outsider dominance" in ownership of the organizations and tour operators that organize and run slum tours. In order to make it more sustainable, there is a need to conduct tours in a more humane manner, ensure that the benefits accrued trickle down to the community and advertising is more respectful. One powerful example is the Santa Marta favela (slum) of Rio de Janeiro, Brazil, where the tour guides have organized themselves as a committee to collectively manage the tourism enterprise and to promote their services as a brand of community-based tourism in opposition to outside commercial tour operators [23].

Another study in Chile proved that tourism can positively help in reducing territorial poverty, as long as strategies that compel the tourism activity to favor the creation of jobs and investments in the slums are employed. This also brings development to the municipalities [24]. Similar studies in Mexico and Ecuador also concluded that the promotion of international tourism as a development strategy helped in poverty reduction [25,26]).

### 2.4. The Slum Tourist

Although some tourists may undertake the slum tour in order to help and support slum dwellers, the vast majority are just driven by curiosity [22]. After all, there is nothing wrong with being curious about how other people live, as it is human nature to learn more

about the unknown, and slum tourism allows a glimpse into an alternate life [7] that most people in the developed world could not imagine possible. Some tourists act as "connectors" to the slum dwellers, influencing the shifts in the political, social, economic, cultural, and material dimensions of the slum area [27], while others are motivated socially and culturally at the transcultural, international levels to investigate exotic types of poverty [28]. Many of the tours amount to simple voyeurism [29]. Tourists enter the slum to immerse themselves in a different environment, moved by urban deprivation, feeling at the same time empathy and solidarity with the slum dwellers as well as a sense of discomfort and distance, while remembering in a visceral way that they do not belong there [30], but while depicting slums as productive cultural spaces, tourists are able to resist the stigma associated with the slum tourism and position themselves as ethical, enlightened, and morally superior beings [31], while some may experience the so-called horrors of capitalism [32].

For many tourists, the destination's poverty seems to be the decision-maker when opting for the tour, while others may try to discover the heritage of the region while visiting the slums [33]. As per Farmaki and Pappas [34], four solutions were found that influenced the slum tourist decision: the cultural influence and poverty of the destination, the destination aspects, the specific poverty issues of the region to be visited, and the travel experience of seeing poverty. However, most of the interviewed volunteer tourists to Brazilian favelas had difficulty justifying how their tours helped the community visited and what improvements could be brought about to the [35]. Another incentive to visit the slums has to do with the way how poverty is marketed as a natural feature of certain tourist destinations in developing or underdeveloped countries, and how low-income communities and lifestyles are commoditized for and by the gaze of the west as an "authentic" adventure experience [36].

*2.5. The Slum Dwellers' View*

Some authors have raised the issue of privacy [5,37] where rich people visit the places where poor people live, in tin and scrap metal houses, that are invaded by tourists, and whether it does not make them feel like caged animals in the zoo [38], that eventually may result in mental trauma for the slum dwellers, especially women and children [37].

In a study in the Rocinha slum of Rio de Janeiro, Brazil, 83% of the slum residents felt tourism was a positive factor [39]. Similar opinions were obtained in studies in Katutura slum, Windhoek, Namibia [40]. A similar study in Turkey's Istanbul slum areas showed that slum dwellers showed a positive attitude towards visitors, and supported tourism, mainly because of the economic gains earned from the tourists as well as for the social interaction [41]. A study on the Dharavi slum of Mumbai city (India) about the perceptions of the residents towards slum tourists identified four perspectives: apprehensive, positive, indifferent, and skeptical. However, over time, the slum dwellers of Dharavi have become less excited about tourists' presence but have not developed a negative attitude towards them. Although some residents criticized the tourists, a lack of knowledge about tourism's contribution to community development projects was observed in the study, but still, slum tourism was not viewed as exploitative [42].

Tourism can be instrumental in alleviating the poverty of the slum dwellers, by providing economic, socio-cultural, and even environmental benefits to an impoverished community, if the strategy makers are sensitized towards the elaboration of effective policies and interventions to develop effective and creative tourism practices that do not only benefit the travel companies but the entire community in a sustainable way [43]. A similar study in Tibet proved that governmental intervention in promoting diverse linkages enabled residents to occupy an active position in the tourism value chain, with improved participation and income in the industry, with the eventual outcome of poverty alleviation [44].

So, whether one likes it or not, slum tourism is becoming popular and is on the rise. It ought to be more regularized, in order to ensure that the benefits percolating to the poor are greater. There are still many tourists that hesitate to visit certain regions or countries

when they perceive the health risks to be high. Policymakers ought to create tailor-made scenarios that offer more prominent sustainable tourism options to the undecided, in order to increase the number of visitors and the revenue [45].

## 3. Methodology

Based on the principles of the Global Code of Ethics for Tourism (GCET) proposed by the UNWTO [13], and with the objective of knowing the attitudes of residents and tourists regarding "slum tourism" in India, namely their perceptions of the positive and negative impacts of this type of tourism, a descriptive study was developed based on a survey [46,47].

In more specific terms, it was intended to (i) know general perceptions about slum tourism in India (ii) assess the benefits of "slum tourism"; (iii) assess the negative impacts of slum tourism; (iv) identify differences in attitudes towards slum tourism as a function of gender and the experience (or not) of visiting a slum; (v) analyze respondents' perceptions of compliance with the "Global Code of Ethics for Tourism".

In the construction of the survey, the studies identified in the literature that analyzed the impacts of slum tourism in countries such as Kenya, Brazil, Mexico, and South Africa were considered [5–7,9,17,39,48,49].

To assess the positive and negative impacts of slum tourism, the scale proposed by Madrigal [48] and Kieti and Magio [7] was used, as well as part of the scale by Mano et al. [50], with the final questionnaire consisting of 21 statements (Table 1).

**Table 1.** Questionnaire statements on Slum Tourism.

| Attitudinal Statements | Items | Authors |
|---|---|---|
| **Positive Statement** | Slum tourism (ST) is a pleasure | Madrigal [48]; Kieti and Magio [7] |
| | Right choice to embrace ST | Madrigal [48]; Kieti and Magio [7] |
| | No future for the area without ST | Madrigal [48]; Kieti and Magio [7] |
| | Area better place to live thanks slum tourism | Madrigal [48]; Kieti and Magio [7] |
| | I support the approval of ST in this area | Madrigal [48]; Kieti and Magio [7] |
| | Resident population receive social benefits from ST | Madrigal [48]; Kieti and Magio [7] |
| | Resident population obtains economic benefits from ST | Madrigal [48]; Kieti and Magio [7] |
| | Overall, all residents benefit from ST | Madrigal [48]; Kieti and Magio [7] |
| | Tourism can have economic advantages to slums and local entrepreneurs, | Mano et al., 2017 [50] |
| | Social projects should be benefited by touristic visits. | Mano et al., 2017 [50] |
| | The interaction between slum residents and tourists positive. | Mano et al., 2017 [50] |
| | Tourism in slums can contribute to local social development | Mano et al., 2017 [50] |
| **Negative Statement** | Hard to accept slum tourism | Madrigal [48]; Kieti and Magio [7] |
| | Not appropriate for this place | Madrigal [48]; Kieti and Magio [7] |
| | It is embarrassing | Madrigal [48]; Kieti and Magio [7] |
| | I don't care if we have slum tourism in this town. | Madrigal [48]; Kieti and Magio [7] |
| | Money goes to outsiders | Madrigal [48]; Kieti and Magio [7] |
| | Many people have moved away | Madrigal [48]; Kieti and Magio [7] |
| | Tourists do not interact with locals | Madrigal [48]; Kieti and Magio [7] |
| | Slum tourism increases human traffic. | Madrigal [48]; Kieti and Magio [7] |
| | ST can bring economic disadvantages (increase in the cost of living and real estate speculation) | Mano et al., 2017 [50] |

Source: Adapted from Madrigal [48], Kieti and Magio [7] and Mano et al. [50].

To analyze the respondents' perception of the commitment to the "Global Code of Ethics for Tourism", the 10 generic codes proposed by the UNWTO [13] were used.

The questionnaire had both closed and open-ended questions administered to 202 residents or tourists. Respondents responded on a 5-point Likert scale, where "1" means "strongly disagree" and "5" means "strongly agree".

The questionnaire was built on "Google Forms" and after carrying out a pre-test [46,47] with 15 visitors, it was made available online between January 2022 and March 2022, to a non-probabilistic convenience sample [47].

## 4. Data Analysis

A total of 237 questionnaires were collected, of which 202 (85%) were validated, which is considered an acceptable number that allows for the analysis and statistical treatment of the data.

To summarize the responses and make assumptions about the survey data, descriptive statistics were created. On the other hand, reliability tests were carried out using Cronbach's Alpha on the scale items to ensure good internal consistency. A factor analysis was also carried out, using the Principal Component Analysis (PCA) method to reduce a large dimension of data to a relatively smaller number of dimensions, components, or latent factors [46,47]). An analysis of variance (Levene test) and comparative means (test-*t*) were conducted to explore possible associations between respondents' gender and slum visitors (or not).

As can be seen in the following Table 2, most respondents (138) belong to the female gender (68.3%) and the rest (64) to the male gender (31.7%).

**Table 2.** Sample.

|  |  | F | % |
|---|---|---|---|
| **Gender** | Male | 138 | 68.3 |
|  | Female | 64 | 31.7 |
| **Age Groups** | 20–30 years old | 118 | 58.4 |
|  | 31–40 years old | 29 | 14.4 |
|  | 41–50 years old | 18 | 8.9 |
|  | 51–60 years old | 33 | 16.3 |
|  | >60 years old | 4 | 2.0 |
| **Qualifications** | High School | 7 | 3.5 |
|  | Bachelor | 83 | 41.1 |
|  | Master | 84 | 41.6 |
|  | Doctorate | 28 | 13.9 |
| **Scientific area** | Business Sciences (Economy, Management, Account, . . . ) | 57 | 28.2 |
|  | Human Sciences (Psychology, Sociology, . . . ) | 66 | 32.7 |
|  | Engineering and technology | 19 | 9.4 |
| **Country** | India | 176 | 87.1 |
|  | Other | 26 | 12.9 |
| **Visited Slum** | Yes | 135 | 66.8 |
|  | No | 67 | 33.2 |

Source: Own study.

Regarding educational qualifications (Table 3) it can be seen that the bulk of the sample has a master's degree (41.6%) and a bachelor's degree (41.1%). Only 29 respondents (13.9%) have a Ph.D. and high school (3.5%). The predominant scientific areas of study for respondents were Human Sciences (32.7%) and Business Sciences (28.2%).

**Table 3.** Attitudes toward Slum Tourism (positive and negative statements).

| Attitudinal Statements | Statement | Strongly Disagree 1 | Disagree 2 | Undecided 3 | Agree 4 | Strongly Agree 5 | M | SD |
|---|---|---|---|---|---|---|---|---|
| | | F (%) | F (%) | F (%) | F (%) | F (%) | | |
| Positive Statement (n = 202) | Slum tourism (ST) is a pleasure | 67 (33.2%) | 37 (18.3%) | 64 (31.7%) | 20 (9.9%) | 14 (6.9%) | 2.39 | 1.234 |
| | Right choice to embrace ST | 46 (22.8%) | 40 (19.8%) | 58 (28.7%) | 41 (20.3%) | 17 (8.4%) | 2.72 | 1.256 |
| | No future for the area without ST | 63 (31.2%) | 52 (25.7%) | 47 (23.3%) | 28 (13.9%) | 12 (5.9%) | 2.38 | 1.225 |
| | Area better place to live thanks slum tourism | 50 (24.8%) | 39 (19.3%) | 60 (29.7%) | 29 (14.4%) | 24 (11.9%) | 2.69 | 1.310 |
| | I support the approval of ST in this area | 48 (23.8%) | 30 (14.9%) | 43 (21.3%) | 47 (23.3%) | 34 (16.8%) | 2.95 | 1.418 |
| | Resident population receive social benefits from ST | 20 (9.9%) | 38 (18.8%) | 69 (34.2%) | 49 (24.3%) | 26 (12.9%) | 3.11 | 1.156 |
| | Resident population obtains economic benefits from ST | 17 (8.4%) | 40 (19.8%) | 60 (29.7%) | 51 (25.2%) | 34 (16.8%) | 3.22 | 1.191 |
| | Overall, all residents benefit from ST | 34 (16.8%) | 43 (21.3%) | 60 (29.3%) | 42 (20.8%) | 23 (11.4%) | 2.89 | 1.243 |
| | Tourism can have economic advantages to slums and local entrepreneurs, | 19 (9.4%) | 30 (14.9%) | 46 (22.8%) | 68 (33.7%) | 39 (19.3%) | 3.39 | 1.221 |
| | Social projects should be benefited by touristic visits. | 12 (5.9%) | 18 (8.9%) | 49 (24.3%) | 68 (33.7%) | 55 (27.2%) | 3.67 | 1.143 |
| | The interaction between slum residents and tourists positive. | 13 (6.4%) | 22 (10.9%) | 77 (38.1%) | 56 (27.7%) | 34 (16.8%) | 3.38 | 1.087 |
| | Tourism in slums can contribute to local social development | 16 (7.9%) | 28 (13.9%) | 55 (27.2%) | 64 (31.7%) | 39 (19.3%) | 3.41 | 1.178 |
| | **Total mean positive statement** | 34 (16.8) | 35 (17.3) | 57 (28.2) | 47 (23.3) | 29 (14.4) | 3.01 | 1.221 |
| Negative statement (n = 202) | Hard to accept slum tourism | 21 (10.4%) | 28 (13.9%) | 50 (24.8%) | 36 (17.8%) | 67 (33.2%) | 3.50 | 1.350 |
| | Not appropriate for this place | 19 (9.4%) | 39 (19.3%) | 55 (27.2%) | 22 (10.9%) | 67 (33.2%) | 3.39 | 1.364 |
| | It is embarrassing | 54 (26.7%) | 33 (16.3%) | 43 (21.3%) | 26 (12.9%) | 46 (22.8%) | 2.89 | 1.507 |
| | I don't care if we have slum tourism in this town. | 42 (20.8%) | 50 (24.8%) | 52 (25.7%) | 32 (15.8%) | 26 (12.9%) | 2.75 | 1.304 |
| | Money goes to outsiders | 12 (5.9%) | 18 (8.9%) | 69 (34.2%) | 44 (21.8%) | 59 (29.2%) | 3.59 | 1.169 |
| | Many people have moved away | 18 (8.9%) | 29 (14.4%) | 73 (36.1%) | 44 (21.8%) | 38 (18.8%) | 3.27 | 1.185 |
| | Tourists do not interact with locals | 20 (9.9%) | 39 (19.3%) | 75 (37.1%) | 39 (19.3%) | 29 (14.4%) | 3.09 | 1.164 |

**Table 3.** *Cont.*

| Attitudinal Statements | Statement | Strongly Disagree 1 | Disagree 2 | Undecided 3 | Agree 4 | Strongly Agree 5 | M | SD |
| --- | --- | --- | --- | --- | --- | --- | --- | --- |
| | | F (%) | F (%) | F (%) | F (%) | F (%) | | |
| | Slum tourism increase human traffic. | 3 (1.5%) | 21 (10.4%) | 68 (33.7%) | 55 (27.2%) | 55 (27.2%) | 3.68 | 1.031 |
| | ST can bring economic disadvantages (increase in the cost of living and real estate speculation) | 17 (8.4%) | 49 (24.3%) | 59 (29.2%) | 50 (24.8%) | 27 (13.4%) | 3.10 | 1.165 |
| | **Total mean negative statement** | 23 (11.4%) | 34 (16.8%) | 60 (29.7%) | 39 (19.3%) | 46 (22.8%) | 3.29 | 1.248 |

Source: Own study.

In terms of country of origin, it appears that the majority of respondents are from India (66.8%) and the rest belong to a very dispersed set of nationalities, so they were aggregated in the category of "Other country" (33.2%).

Of the total number of respondents, 135 have already visited the "Slums" (66.8%) and only 67 (33.2%) have not visited.

The scale shows good internal consistency, having obtained a Cronbach's Alpha of 0.711 for all the 21 items that make up the scale.

As can be seen in the table below (Table 2), and in global terms, negative perceptions about "Slum tourism" are higher (M = 3.38; SD = 1.48) than positive perceptions (M = 3.01; SD = 1.221).

Regarding the "positive statements" about "slum tourism" there were two items with scores below the arithmetic mean (M < 2.5). These are the items "No future for the area without ST" (M = 2.38; SD = 1.225) and "Slum tourism (ST) is a pleasure" (M = 2.39; SD = 1.234). However, most respondents consider that "Social projects should be benefited by touristic visits" (M = 3.67; SD = 1.143) and "Tourism in slums can contribute to local social development" (M = 3.41; SD = 1.178).

As for the "negative statement", all items obtained agreement values above the arithmetic mean (M = 3.29; SD = 1.248), especially the evaluation of the items "Slum tourism increase human traffic" (M = 3.68; SD = 1.031) and "Money goes to outsiders" (M = 3.59; SD = 1.169).

In order to assess the dimensionality of the scale used and reduce and group the number of correlated variables, exploratory factor analysis was carried out using the Principal Component Analysis (PCA) method, which allowed the identification of six factors that explain 70.3% of the variance (Table 4). The Kaiser criteria (eigen values greater than 1) and the "scree plot" generate credible solutions for choosing the number of factors to retain. The Kaiser–Meyer–Olkin is high (KMO = 0.876), with the Bartlett test having a significance level of $p = 0.000$, allowing the continuation of the factor analysis.

The communalities (proportion of the variance of each variable explained by the principal components) show a strong relationship with the retained factors, all being values greater than 50%.

Given the existence of six factors, the varimax rotation was performed, having obtained the following alignment of the factors that became known as (Table 5): Component 1: General Benefits; Component 2: Best Solution; Component 3: Personal Shame; Component 4: Cost and Damages; Component 5: Economic disadvantages; Component 6: Indifference

**Table 4.** Results for factor analysis (Principal Component Analysis—PCA).

| Component | Initial Own Values | | | Square Extraction Sums | |
|---|---|---|---|---|---|
| | Total | % of Variance | % Cumulative | Total | % Variance |
| 1 | 7.518 | 35.802 | 35.802 | 7.518 | 35.802 |
| 2 | 2.190 | 10.428 | 46.231 | 2.190 | 10.428 |
| 3 | 1.585 | 7.548 | 53.778 | 1.585 | 7.548 |
| 4 | 1.313 | 6.254 | 60.032 | 1.313 | 6.254 |
| 5 | 1.122 | 5.341 | 65.373 | 1.122 | 5.341 |
| 6 | 1.042 | 4.962 | 70.336 | 1.042 | 4.962 |
| 7 | 0.710 | 3.383 | 73.719 | | |
| 8 | 0.681 | 3.244 | 76.962 | | |
| 9 | 0.620 | 2.955 | 79.917 | | |
| 10 | 0.566 | 2.697 | 82.614 | | |
| 11 | 0.527 | 2.508 | 85.122 | | |
| 12 | 0.458 | 2.181 | 87.303 | | |
| 13 | 0.419 | 1.996 | 89.299 | | |
| 14 | 0.382 | 1.820 | 91.119 | | |
| 15 | 0.361 | 1.719 | 92.838 | | |
| 16 | 0.334 | 1.591 | 94.429 | | |
| 17 | 0.281 | 1.340 | 95.769 | | |
| 18 | 0.258 | 1.227 | 96.996 | | |
| 19 | 0.232 | 1.106 | 98.102 | | |
| 20 | 0.218 | 1.036 | 99.138 | | |
| 21 | 0.181 | 0.862 | 100.000 | | |

| | | |
|---|---|---|
| Kaiser–Meyer–Olkin measure (KMO) of sample adequation | | 0.876 |
| | Chi-square approximation | 2080.063 |
| Bartlett spherical test | Df | 210 |
| | Sig. | 0.000 |

Extraction Method: Principal Component Analysis. 6 components extracted. Source: Own study.

**Table 5.** Rotating component matrix a.

| Statements | Components | | | | | |
|---|---|---|---|---|---|---|
| | General Benefits | Best Solution | Personal Shame | Costs and Damages | Economic Disadvantages | Indifference |
| Social projects should be benefited by touristic visits. | 0.784 | | | | | |
| Tourism can have economic advantages to slums and local entrepreneurs, such as job and income creation. | 0.772 | | | | | |
| Resident population receive social benefits from slum tourism and improved quality of life. | 0.753 | | | | | |
| Tourism in slums can contribute to local social development through the organization of their residents. | 0.752 | | | | | |
| The resident population obtains economic benefits from slum tourism (income, employment). | 0.747 | | | | | |
| The interaction between slum residents and tourists positive. | 0.718 | | | | | |
| Overall, all residents benefit from slum tourism in this area. | 0.667 | | | | | |

**Table 5.** *Cont.*

| Statements | Components | | | | | |
|---|---|---|---|---|---|---|
| | General Benefits | Best Solution | Personal Shame | Costs and Damages | Economic Disadvantages | Indifference |
| Without slum tourism this area would have no future. | 0.749 | | | | | |
| Slum tourism has made this area a better place to live. | 0.742 | | | | | |
| This area made the right choice to embrace slum tourism. | 0.713 | | | | | |
| Having slum tourism in this place is a pleasure | 0.663 | | | | | |
| If we had it to do over again, I would support approval of slum tourism in this area. | 0.617 | | | | | |
| Slum tourism is not appropriate for this place | | | 0.855 | | | |
| It is hard for me to accept slum tourism | | | 0.781 | | | |
| I am embarrassed that I live in a community associated with slum tourism. | | | 0.749 | | | |
| Most of the money from slum tourism in this area goes to outsiders. | | | | 0.770 | | |
| Many people have moved away from this area because of slum tourism. | | | | 0.766 | | |
| Slum tourism increase human traffic. | | | | 0.622 | | |
| Slum tourists do not interact with the local residents. | | | | 0.621 | | |
| Slum tourism can bring economic disadvantages to the slums, such as an increase in the cost of living and real estate speculation. | | | | | 0.897 | |
| I don't care if we have slum tourism in this town. | | | | | | 0.931 |

Extraction Method: Principal Component Analysis. Rotation Method: Varimax with Kaiser Normalization. a. Rotation converged in 6 interactions. Source: Own study.

The six factors obtained present weak correlation coefficients (<0.5) and, in some cases, negative (Table 6). Thus, in concrete terms, the factor "General Benefits" is positively and significantly correlated with "Best Solution (r = 0.461; $p$ = 0.000) and with "Costs and Damages" (r = 0.166; $p$ = 0.000). However, it has negative but significant correction coefficients, with "Personal Shame" (r = −0.351; $p$ = 0.000) and "Economic Disadvantages" (r = −0.222; $p$ = 0.000).

**Table 6.** Correlations between variables.

| | GB | BS | PS | ED | CD | IN |
|---|---|---|---|---|---|---|
| General Benefits (GB) | - | 0.461 ** | −0.351 ** | −0.222 ** | 0.166 * | 0.104 |
| Best Solution (BS) | | - | −0.177 * | −0.069 | 0.116 | 0.129 |
| Personal Shame (PS) | | | - | 0.392 ** | 0.046 | 0.007 |
| Costs and Damages (CD) | | | | - | 0.073 | −0.012 |
| Economic disadvantages (ED) | | | | | - | 0.007 |
| Indifference (IN) | | | | | | - |

Source: Own study. **. Correlation is significant at level 0.01 (2 extremities). *. Correlation is significant at level 0.05 (2 extremities).

As can be seen in the Table 7 we can assume the equality of variances for both genders in terms of the factor "Personal Shame" (F = 7.711, $p$-value = 0.006) and "Costs and Damages" (F = 4.408; $p$-value = 0.066).

**Table 7.** Compared by gender.

| Factor | Gender | Descriptive | | | Levene Test for Equality of Variances | | *t*-Test for Equality of Averages | | | |
|---|---|---|---|---|---|---|---|---|---|---|
| | | N | M | SD | F | Sig. | t | Sig. | Mean Difference | Difference Standard Error |
| General Benefits | Male | 64 | 2.39 | 1.163 | 0.654 | 0.420 | −0.440 | 0.660 | −0.073 | 0.166 |
| | Female | 138 | 2.46 | 1.068 | | | −0.427 | 0.671 | −0.073 | 0.171 |
| Best Solution | Male | 64 | 1.52 | 0.891 | 2.058 | 0.153 | −2.677 | **0.008** | −0.390 | 0.146 |
| | Female | 138 | 1.91 | 0.996 | | | −2.788 | **0.006** | −0.390 | 0.140 |
| Personal Shame | Male | 64 | 2.63 | 1.589 | 7.711 | **0.006** | 0.383 | 0.702 | 0.082 | 0.213 |
| | Female | 138 | 2.54 | 1.313 | | | 0.358 | 0.721 | 0.082 | 0.228 |
| Costs and Damages | Male | 64 | 2.64 | 1.200 | 3.408 | **0.066** | 0.779 | 0.437 | 0.126 | 0.162 |
| | Female | 138 | 2.51 | 1.005 | | | 0.730 | 0.467 | 0.126 | 0.173 |
| Economic disadvantages | Male | 64 | 3.00 | 1.272 | 0.615 | 0.434 | −0.863 | 0.389 | −0.152 | 0.176 |
| | Female | 138 | 3.15 | 1.113 | | | −0.822 | 0.413 | −0.152 | 0.185 |
| Indifference | Male | 64 | 2.75 | 1.403 | 0.587 | 0.444 | −0.018 | 0.985 | −0.004 | 0.198 |
| | Female | 138 | 2.75 | 1.260 | | | −0.018 | 0.986 | −0.004 | 0.206 |

Source: Own study. The significant values (sig) are shown in bold.

Through the *t*-test to compare averages, we found that there are significant differences between genders (*p*-value = 0.008) in terms of the "Best solution" factor, and this assessment is higher in men (M = 1.91) than in women (M = 1.52).

In the Table 8 we can assume the equality of variances for two genders at the level of the factor "Costs and Damages" (F = 5.6311, *p*-value = 0.019).

**Table 8.** Comparison of "visitor and non-visitor".

| Factor | Have You Ever Visited Slums | Descriptive | | | Levene Test for Equality of Variances | | *t*-Test for Equality of Averages | | | |
|---|---|---|---|---|---|---|---|---|---|---|
| | | N | M | Sig. | F | Sig. | t | Sig. | Mean Difference | Difference Standard Error |
| General Benefits | Yes | 135 | 2.36 | 1.096 | 0.685 | 0.409 | −1.570 | 0.118 | −0.256 | 0.163 |
| | No | 67 | 2.61 | 1.086 | | | −1.575 | 0.118 | −0.256 | 0.163 |
| Best Solution | Yes | 135 | 1.62 | 0.929 | 1.017 | 0.314 | −3.383 | **0.001** | −0.482 | 0.143 |
| | No | 67 | 2.10 | 1.002 | | | −3.298 | **0.001** | −0.482 | 0.146 |
| Personal Shame | Yes | 135 | 2.56 | 1.433 | 1.128 | 0.289 | −0.091 | 0.928 | −0.019 | 0.210 |
| | No | 67 | 2.58 | 1.350 | | | −0.093 | 0.926 | −0.019 | 0.206 |
| Costs and Damages | Yes | 135 | 2.53 | 1.138 | 5.631 | **0.019** | −0.537 | 0.592 | −0.086 | 0.160 |
| | No | 67 | 2.61 | 0.920 | | | −0.577 | 0.565 | −0.086 | 0.149 |
| Economic disadvantages | Yes | 135 | 3.07 | 1.195 | 0.874 | 0.351 | −0.517 | 0.523 | −0.125 | 0.195 |
| | No | 67 | 3.16 | 1.109 | | | −0.530 | 0.521 | −0.125 | 0.194 |
| Indifference | Yes | 135 | 2.71 | 1.315 | 0.064 | 0.800 | −0.639 | 0.606 | −0.090 | 0.174 |
| | No | 67 | 2.84 | 1.286 | | | −0.644 | 0.597 | −0.090 | 0.170 |

Source: Own study. The significant values (sig) are shown in bold.

Through the *t*-test to compare averages, we found that there are significant differences between genders (*p*-value = 0.001) in terms of the "Best solution" factor, and this evaluation is higher in non-visitors (M = 2.10) than in visitors (M = 1.62).

The scale used to measure the 10 principles of the Global Code of Ethics for Tourism (GCET) had a good internal consistency (Cronbach's Alpha of 0.702).

Respondents' perceptions of compliance with the 10 ethical principles of tourism in the region are quite different (Table 9), although, overall, the general average has revealed a positive result (M = 3.051)

**Table 9.** Global Code of Ethics for Tourism (GCET).

| Global Code of Ethics for Tourism (GCET) Alpha Cronbach: 0.0702 | Strongly Disagree 1 | Disagree 2 | Undecided 3 | Agree 4 | Strongly Agree 5 | M | SD |
|---|---|---|---|---|---|---|---|
| | F (%) | F (%) | F (%) | F (%) | F (%) | | |
| 1."Tourism must contribute to mutual understanding and respect between people and society" | 19 (9.4%) | 30 (14.9%) | 46 (22.8%) | 68 (33.7%) | 39 (19.3%) | **3.82** | 0.908 |
| 2."Tourism should be a vehicle for individual and collective fulfillment" | 63 (31.2%) | 52 (25.7%) | 47 (23.3%) | 28 (13.9%) | 12 (5.9) | 2.96 | 1.134 |
| 3."Tourism is a factor for sustainable development" | 20 (9.9%) | 38 (18.8%) | 69 (34.2%) | 49 (24.3%) | 26 (12.9%) | 2.94 | 1.114 |
| 4."Tourism is a user of the cultural heritage of mankind and a contributor for its enhancement" | 50 (24.8%) | 39 (19.3) | 60 (29.7) | 29 (14.4%) | 24 (11.9%) | 2.69 | 1.310 |
| 5."Tourism should be a beneficial activity for the host country and its communities" | 34 (16.8%) | 43 (21.3%) | 60 (29.7%) | 42 (20.8%) | 23 (11.4%) | **3.16** | 1.158 |
| 6."Stakeholders' have obligations in tourism development" | 12 (5.9%) | 18 (8.9%) | 49 (24.3%) | 68 (33.7%) | 55 (27.2%) | **3.67** | 1.143 |
| 7."Tourism has rights" | 42 (20.8%) | 50 (24.8%) | 52 (25.7%) | 32 (15.8%) | 26 (12.9%) | 2.75 | 1.304 |
| 8."There should be liberty and freedom of tourists' movements" | 51 (25.2%) | 50 (24.8) | 68 (33.7%) | 27 (13.4%) | 6 (3.0%) | 2.54 | 1.120 |
| 9."Workers and entrepreneurs in the tourism industry have rights" | 107 (53%) | 46 (22.8%) | 37 (18.3% | 10 (5%) | 2 (1%) | 1.82 | 1.013 |
| 10."The global code of ethics for tourism should be implemented by every country" | 21 (10.4%) | 28 (13.9%) | 50 (24.8%) | 36 (17.8%) | 67 (33.2%) | **4.16** | 0.895 |
| **Total mean (GCET)** | 42 (20.8%) | 39 (19.3%) | 54 (26.7%) | 39 (19.3%) | 28 (13.9%) | 3.051 | 1.109 |

Source: Own study. The highest "Mean" values are shown in bold.

The indicators that obtained the highest scores (compliance with the Global Ethical Codes for Tourism) were the following: "The global code of ethics for tourism should be implemented by every country" (M = 4.16; SD = 0.995) with 51% agreement; "Tourism must contribute to mutual understanding and respect between people and society" (M = 3.82; SD = 0.908); "Stakeholders' have obligations in tourism development" (M = 3.67; SD = 1.143) and "Tourism should be a beneficial activity for the host country and its communities" (M = 3.16; SD = 1.158).

However, the results are worrying regarding the items "Workers and entrepreneurs in the tourism industry have rights" (M = 1.82; SD = 1.1013); "There should be liberty and freedom of tourists' movements" (M = 2.54; SD = 1.120) and "Tourism is a user of the cultural heritage of mankind and a contributor for its enhancement" (M = 2.69; SD = 1.310).

In the end, the respondents expressed their opinion about "Slum Tourism", so below are some relevant statements.

Thus, some respondents consider that some tourists like to visit these places as spectators (voyeurism), but that this does not bring benefits to communities and places ("Based on my experience, many bloggers just come, film, and leave. No benefit is given to the

resident, they are just made the subject of spectator") considering that they do not support this type of tourism ("I don't support slum tourism").

They consider this reality inhuman and undignified ("I believe every human has a right to live with dignity and slum tourism is violation of this right to dignity") and that there are people and economic agents taking advantage of these conditions of poverty and misery of people to make money ("Slum tourism is a distorted concept that seeks to monetize poor living conditions. It tried to accommodate human empathy in economic framework robbing it of the subtle subjective realities of places like slums").

Some even warn that this tourist activity can be misleading as it can perpetuate this situation over time ("Slum tourism is very much new to me. However, I believe this would only glorify slums and people residing there . . . making it even more difficult to persuade them to relocate"), making it difficult to leave these places and relocate them.

However, there are some respondents who consider that slum tourism can be beneficial insofar as it can improve the living conditions of these populations ("This will improve the life of people in slum area") and by knowing the reality of the country ("Visiting modern day life spaces is the true essence of knowing any country") allows the development of policies and actions to improve the living conditions of these populations ("Slum is undesirable reality of the 21st century, especially in India. Through concerted actions, it is possible to eliminate such inhumane way of living").

## 5. Discussion

This study aimed to understand the attitudes of residents of the city about slum tourism in Dharavi, Mumbai, India, seeking to understand its positive and negative impact.

According to Jaffe et al., [30] tourists seek to experience different environments, try to gain empathy and solidarity with the residents, as well as to discover new cultural spaces. Our survey revealed that respondents are aware of the problems and challenges arising from slum tourism in India but are divided on attitudes toward slum tourism given that perceptions of negative impacts (M = 3.29) are higher than perceptions of positive impacts (M = 3.01). Regarding the "positive statements" about "slum tourism" the majority of respondents consider that social projects should be benefited from touristic visits (M = 3.67) and tourism in slums can contribute to local social development (M = 3.41). In turn, regarding the "negative statement", major of respondents consider that slum tourism increases "human traffic" (M = 3.68) and "money goes to outsiders" (M = 3.59).

Exploratory factor analysis using Principal Component Analysis (PCA) method revealed that there are six factors that explain this phenomenon: General Benefits, Best Solution, Personal Shame, Costs and Damages, Economic Disadvantages, and Indifference.

The study revealed that there are statistically significant differences between genders regarding the "Best Solution" dimension, and this assessment is higher in men (M = 1.91) than in women (M = 1.52). Likewise, we found statistically significant differences between "visitors and non-visitors" ($p$-value = 0.001) in relation to the "Best Solution" dimension, and this evaluation is higher in non-visitors (M = 2.10) than in visitors (M = 1.62).

Some of the tourists visit "slums" as spectators and, in this sense, they do not directly contribute to the well-being or benefit of residents in these regions. As mentioned in the study of slum tourism in Kenya, Kieti and Magio [7] proved that the benefits of slum tourism were insignificant to make residents support its development, we also found in this research that tourism in slums was insignificant.

As evidenced in the literature [7,22,28,29,33,34], our investigation confirmed that the perceived negative impacts of slum tourism are greater than the expected benefits, with few opportunities for slum residents. Chhabra and Chowdury [29] conducted a pilot study on Indian slums and concluded that there were heavy traces of voyeurism on the part of the tourists. Kieti and Magio [7] studied Kenyan slums where they found a higher level of negative behavior as compared to a positive one, given that the slum residents feel that the benefits of this kind of tourism do not benefit them.

Similar to the studies identified in the literature, most tourists visit the slums out of curiosity, "to see poverty", to know a different reality, and for simple voyeurism [28,29]. Likewise, the study by Frenzel et al. [22] also found that many tourists simply undertake the tour to satisfy their curiosity and do not contribute to the area development, but the tour operators earn more with this activity. So, in terms of balance, and in line with the studies identified in the literature, our study reveals that "slum tourism" presents greater harm than benefits for the locality and the resident population. In addition, the recognized benefits are mostly for external agents, namely for tour operators.

In fact, this invasion of the territory, where the "rich visit the places where the poor live", evidences a distance between residents and visitors, which, as international studies mention [5,37] can cause mental and social trauma in slum residents who feel watched, caged, and excluded. The label "slum" does not promote the dignity of residents and their struggle for more dignified lives and the desire for equal opportunities [6].

The experience of visiting the slum, as a cultural and social space, and the "sense of the place" lived "in loco" allows some visitors to position themselves as enlightened beings about the socio-economic reality of the populations and to know the consequences of capitalism and from "dual" growth (development and underdevelopment) of the territory [30–32].

Despite the results of our study indicating that the perceptions of the negative aspects are superior to the expected benefits, many respondents recognize the advantages of this type of tourism, as mentioned in similar studies [39–41,43–45]. In fact, respondents showed some apprehension regarding slum tourism, but they have a positive attitude towards tourists and support this activity due to the economic gains obtained and the benefits of social interaction. There are even some respondents who are indifferent to this type of tourism, which confirms the results of the study by Mano et al., [50], and Guzel et al., [41].

In a complementary way, this study demonstrates that slum tourism, according to some of the respondents, is the realization of an inhuman reality, with economic agents taking advantage of this situation to obtain economic benefits.

In this sense, and based on compliance with the 10 principles of the "Global Code of Ethics for Tourism" (1999) [13], the data reveal that:

1.  "Tourism must contribute to mutual understanding and respect between people and society"

Slum tourists may be voyeuristic to some extent, but none of them intend to offend or disrespect the resident populace. The aim of the majority of tourists is to see the "unknown" and live a new experience. However, the way slum tourism is conducted in Dharavi, may not respect the privacy of the slum dwellers and may be stressful for some women and children. Nevertheless, it is a fact that tourism contributes to the local economy and to some extent, to the welfare of the residents. Although it does not qualify entirely as a sustainable activity, the existence of slum tourists helps directly and indirectly in the development of the area, mainly because of the money that tourists spend with the locals and traditional businesses that have been established in the slum area.

2.  "Tourism should be a vehicle for individual and collective fulfillment"

Many tourists come to Dharavi in order to satisfy their curiosity. As per Kieti and Magio [7], there is nothing wrong with being curious and visiting something that is unknown to us, in order to fulfill our wish. Some have deep-rooted concerns for the poor and destitute and come to experience a way of living that is not possible to feel in the home country.

3.  "Tourism is a factor for sustainable development"

Tourism, no doubt, is a beneficial activity for India, with Mumbai's luxury hotels and heritage properties earning millions with tourism, but whether as stakeholders, tourists are obliged to help in tourism development remains undefined in this study. However, the issue as to whether the benefits of such development percolate to the slum dwellers,

remains unanswered, given the fact that the sector is fragmented and there is at present only one tour operator conducting such visits.

4. "Tourism is a user of the cultural heritage of mankind and a contributor for its enhancement"

Tourists come to experience the cultural heritage of India and may to some extent contribute to its enhancement, by being goodwill ambassadors in their countries of origin, upon their return, resulting in the increased curiosity and interest of more people to participate in the experience.

5. "Tourism should be a beneficial activity for the host country and its communities"

The benefits of tourism are many, covering not only travel services and boarding-lodging activities, but a wide range of independent but related sectors like transport, accommodation, food and beverage, and entertainment, among others [1]. However, which communities of India benefit from it remains unanswered, especially if the focus is on the slum dwellers. Nevertheless, many slum dwellers manage to make a fast buck by helping tourists as baggage carriers, cleaners, and touts, or even by opening small businesses inside the slum area, like eating joints or mini shops selling trinkets or [19].

6. "Stakeholders' have obligations in tourism development"

Although the definition of stakeholders is vague, it is implied that they do have obligations to develop the sector in which they operate. The same is true of slum tourism stakeholders, but whether all of them contribute to the development of the sector remains unanswered.

Successful tourism development depends greatly on excellent cooperation and communication between all stakeholders involved in the system [51]. In the field of tourism, relationships and collaborations among various stakeholders can be crucial for long-term sustainability, competitiveness, or even survival in terms of destination competitiveness as well as at the level of individual tourism projects [52].

7. "Tourism has rights"

Community participation has proved to be successful for development in the western world, but in developing countries like India, some operational, structural, and cultural barriers may be there [53]. Anyone can participate in slum tourism, but the curious fact is that there is only one operator in the segment in Dharavi. With the increase in the popularity of this niche type of tourism, probably more companies may enter the segment.

8. "There should be liberty and freedom of tourists' movements"

Any kind of ban or sanction imposed by a country would affect severely its tourism industry [54]. Such bans are mostly seen in communist and totalitarian regimes, where tourists' movements are monitored, and they are not allowed to mingle with the locals. In India a free country, tourists have the right to participate in tourism and have liberty of movement, unlike many other countries in Asia and Africa. There is no restriction on the dress code of the tourist nor any ban on tourism activity in India.

9. "Workers and entrepreneurs in the tourism industry have rights"

Although workers and entrepreneurs' rights depend on the labor laws of the country, certain migrant workers and slum dwellers are severely prejudiced and exploited by ruthless employers, due to the availability of excessively low-paid and low-skilled precarious jobs in tourism and hospitality and the abundance of desperate illegal and poor migrants [55]. Although workers in the organized sector have rights, the same cannot be said about the self-employed touts who try to serve tourists in order to earn a few bucks, a sad fact that can be seen in Indian cities and not only in the slum areas of Mumbai.

10. "The global code of ethics for tourism should be implemented by every country"

The GCET has been the object of several favorable and antagonistic pronouncements. Although there are positive opinions regarding its openness and calls for the responsibility of every stakeholder, it is criticized by others due to its generic nature, lack of

substance, and minimal attention to important issues like the impact of tourism on the environment, etc. [56].

## 6. Conclusions

This article was an in-depth study of the situation of slum tourism in Dharavi, Mumbai's, and Asia's, largest slum area, based on the inquiries conducted via questionnaires given to city residents, in order to understand their opinion about this kind of niche tourism. Although slums are a part of every Indian city, this is the only city where slum tourism is taken as an economic activity, so our study concentrated on it. At this stage, it was not possible to compare Mumbai with other Indian cities.

Slums are a reality of the Indian culture and depict the reality of the millions of poor people who migrate to the cities in search of better living conditions. Most of the residents of the city accept this as a normal fact, as it is probably as old as the city itself. Although some find it unethical, no one would imagine living in Mumbai without Dharavi.

We tried to understand the applicability of the 10 GCET principles in this slum area and, based on our study, we can say that despite the fact the slum tourism in Dharavi does not fulfill completely all the ten principles of GCET and there are many people, both residents, and non-residents, who feel that such a kind of tourism is demeaning to human nature, overall, the majority feel that it is helpful in uplifting the masses and benefits one and all directly, or indirectly. So, whether everyone likes it or not, it is there to stay and should develop and expand over time.

Much remains to be done, as far as the GCET principles are concerned. It will be necessary for the government of India to implement these principles as a regulation rather than a voluntary compliance recommendation, in order to improve the plight of the slum dwellers and many of the operators, like the casual workers who have no rights at present. Eventually, such activity is bound to attract more companies, and then probably the GCET principles may be enforced. The more it develops, the greater would be the wealth that would percolate to the slum dwellers, in the form of employment, entrepreneurship (opening small businesses in Dharavi), etc.

## 7. Limitations and Suggestions for Further Study

This study was quite limited as it simply covered the opinions of the Mumbai city residents (and not slum dwellers) and a few of the tourists and did not cover the other Indian cities, as they do not have any kind of slum tourism activities. Another study could be conducted involving only the slum tourists in order to understand better their opinions before and after the tour. Conversely, a study of the slum dwellers of Dharavi could be undertaken to measure the level of satisfaction they have with this kind of odd business.

It would be interesting to expand this study to other cities in India, in order to compare the plight of slum dwellers and to find out what is their opinion about such a kind of tourism that may eventually start, before expanding it to other Asian/African/Latin American countries, for a more global comparison.

We analyzed the GCET principles based on the responses to our questionnaires and not from the perspective of the other stakeholders (government, economic operators, slum dwellers, tourists, etc.). Future studies could be conducted qualitatively in order to gather the opinion of other specialists and agents, in order to deepen the knowledge and the applicability of the GCET principles.

**Author Contributions:** Conceptualization, A.C., M.S.P., A.d.S. and N.S.; methodology, A.C.; software, A.C.; validation, A.C., M.S.P., A.d.S., N.S., J.F. and I.O.; formal analysis, A.C., M.S.P. and N.S.; investigation, N.S.; resources, A.d.S. and A.C.; data curation, A.d.S. and M.S.P.; writing—original draft preparation, A.C.; writing—review and editing, A.d.S. and N.S.; visualization, A.C., M.S.P., A.d.S., N.S., J.F. and I.O.; supervision, A.C.; project administration, A.C., M.S.P., A.d.S. and N.S. All authors have read and agreed to the published version of the manuscript.

**Funding:** This research received no external funding.

**Institutional Review Board Statement:** The study was conducted in accordance with the Declaration of Helsinki and approved by the Institutional Review Board.

**Informed Consent Statement:** Informed consent was obtained from all subjects involved in the study.

**Data Availability Statement:** Not applicable.

**Conflicts of Interest:** The authors declare no conflict of interest.

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
