# Peer review of "Attitudes towards Slum Tourism in Mumbai, India: Analysis of Positive and Negative Impacts"

_sustainability, doi:10.3390/su141710801_

Round 1

Reviewer 1 Report

At the beginning of section 2.1, a text seems to have been copied that does not correspond. 

I believe that the suitability of the sample should be better justified. Why is a total of 237 questionnaires considered acceptable?

There should be more analysis and interpretation of the survey results. In their current formulation, the conclusions do not seem to follow from the research findings, rather they seem to reproduce opinions of the researchers. 

Author Response

At the beginning of section 2.1, a text seems to have been copied that does not correspond. 

CORRECTED

I believe that the suitability of the sample should be better justified. Why is a total of 237 questionnaires considered acceptable?  

Out of 237 questionnaires, a few were incomplete and only 220 were validated. This sample size of 220 is considered valid as the sample dimension (non-probabilistic) is good enough to enable a statistical analysis of data. In statistical terms, only 195 questionnaires sample would be sufficient for an error term of 7% in a 95% confidence level.

There should be more analysis and interpretation of the survey results. In their current formulation, the conclusions do not seem to follow from the research findings, rather they seem to reproduce opinions of the researchers. 

CORRECTED

Reviewer 2 Report

Please check your dictionary, is the word “slumdweller” connected or separated? In the Keywords section you write it as “slumdweller”, but in the Literature Review section, you write it as “slum dweller”

In the introduction, you should explain in detail, why choose slum tourism as the subject. Why don't you choose another type of uncommon tourism? For example why you did not choose dark tourism. Explain in detail why slum tourism is very important to research

Explain in the Introduction section why you made Dharavi your object of research, and why not other slum areas.

Your title seems very general. Even though you only explore one tourism area. Therefore it is better to narrow the title and focus on Dharavi. This of course shows that your contribution is very narrow. Therefore, in order to make your contribution big enough, you should cover several areas of the slum in India. Not just one area. Use a valid statistical method to choose which slum area to sample.

The first three paragraphs of The Beginning sub-section seem completely out of touch with your research. Impressed you are just adding useless paragraphs

Avoid a paragraph that consists of only one sentence. For example on Page 2 Line 85, there is only one sentence in the paragraph. It's a good idea to check and correct all other paragraphs.

Sentences on lines 100, 101, 102, and 103 seem to be just replications and paraphrases of sentences on lines 34, 35, 36, 37, and 38. It's best to avoid repetitive sentences. Very uncomfortable to read

You have to explain, where each statement instrument in the questionnaire comes from? If you paraphrase from another source, then you should include the citation in each statement. For example, where does the statement “Slum tourism (ST) is a pleasure” come from? So every statement must be clearly sourced by academic reasons. Because there may be other positive or negative statements that have not been accommodated. For example, regarding negative statements, the statement “Slum Tourism can damage local culture” should also be included

Also, some statements seem to mean the same thing. For example, the statement “Resident population obtains economic benefits from ST” and the statement “Tourism can have economic advantages to slums and local entrepreneurs” have the same meaning that is about economical benefits. You must explain why the two things should be separated.

Add a Discussion Chapter that provides an in-depth interpretation of your research results. Compare with the results of previous studies. Give reasonable insight. Provide some future direction. Also, explain the advantages of your research results compared to previous studies. The discussion section is the most important part of a scientific paper because it shows the depth and breadth of the researcher in understanding the research topic. At this time, it seems that your paper is very under-contributed because this section does not exist

You have to explain that with the various negative impacts, then how to reduce or eliminate these negative impacts.

From the start, you said you would analyze based on CGET, but in fact, it only appears at the conclusion. And the contents seem just your personal opinion. Each of these GCET points should be discussed with valid and complete data. Should not be written in the Conclusion section, but discussed in the results chapter and discussion chapter.

Page 1 Line 22, “….like slum tourism, dark tourism, sex tourism”. It's better to add “and” before the “sex tourism” phrase.

In the abstract, please make this sentence more concise. I did not understand the current form

“This paper analyzes the case of slum tourism to Dharavi, India´s commercial capital and largest city and its benefits and drawbacks, as well as the opinion of the common people and slum-dwellers, while observing if the ten principles of “Global Code of Ethics for Tourism” (GCET) has been fulfilled in the country”

Author Response

Please check your dictionary, is the word “slumdweller” connected or separated? In the Keywords section you write it as “slumdweller”, but in the Literature Review section, you write it as “slum dweller”

CORRECTED

In the introduction, you should explain in detail, why choose slum tourism as the subject. Why don't you choose another type of uncommon tourism? For example why you did not choose dark tourism. Explain in detail why slum tourism is very important to research

CORRECTED

Explain in the Introduction section why you made Dharavi your object of research, and why not other slum areas.

CORRECTED

Your title seems very general. Even though you only explore one tourism area. Therefore it is better to narrow the title and focus on Dharavi. This of course shows that your contribution is very narrow. Therefore, in order to make your contribution big enough, you should cover several areas of the slum in India. Not just one area. Use a valid statistical method to choose which slum area to sample.

CORRECTED

The first three paragraphs of The Beginning sub-section seem completely out of touch with your research. Impressed you are just adding useless paragraphs

CORRECTED

Avoid a paragraph that consists of only one sentence. For example on Page 2 Line 85, there is only one sentence in the paragraph. It's a good idea to check and correct all other paragraphs.

CORRECTED

Sentences on lines 100, 101, 102, and 103 seem to be just replications and paraphrases of sentences on lines 34, 35, 36, 37, and 38. It's best to avoid repetitive sentences. Very uncomfortable to read

THERE IS NO SUCH REPETITION

You have to explain, where each statement instrument in the questionnaire comes from? If you paraphrase from another source, then you should include the citation in each statement. For example, where does the statement “Slum tourism (ST) is a pleasure” come from? So every statement must be clearly sourced by academic reasons. Because there may be other positive or negative statements that have not been accommodated. For example, regarding negative statements, the statement “Slum Tourism can damage local culture” should also be included

CORRECTED

Also, some statements seem to mean the same thing. For example, the statement “Resident population obtains economic benefits from ST” and the statement “Tourism can have economic advantages to slums and local entrepreneurs” have the same meaning that is about economical benefits. You must explain why the two things should be separated.

The survey was built based on scales tested and developed by different authors (Madrigal, 1993; kieti & Magio ,2013; and Mano et al, 2017). As these are multidimensional scales and contain latent variables, different items/statements are used (some of which are similar, its normal) that allow the assessment of the same dimension

Add a Discussion Chapter that provides an in-depth interpretation of your research results. Compare with the results of previous studies. Give reasonable insight. Provide some future direction. Also, explain the advantages of your research results compared to previous studies. The discussion section is the most important part of a scientific paper because it shows the depth and breadth of the researcher in understanding the research topic. At this time, it seems that your paper is very under-contributed because this section does not exist

CORRECTED AND INCLUDED

You have to explain that with the various negative impacts, then how to reduce or eliminate these negative impacts.

CORRECTED

From the start, you said you would analyze based on CGET, but in fact, it only appears at the conclusion. And the contents seem just your personal opinion. Each of these GCET points should be discussed with valid and complete data. Should not be written in the Conclusion section, but discussed in the results chapter and discussion chapter.

CORRECTED

Page 1 Line 22, “….like slum tourism, dark tourism, sex tourism”. It's better to add “and” before the “sex tourism” phrase.

CORRECTED

In the abstract, please make this sentence more concise. I did not understand the current form:

“This paper analyzes the case of slum tourism to Dharavi, India´s commercial capital and largest city and its benefits and drawbacks, as well as the opinion of the common people and slum-dwellers, while observing if the ten principles of “Global Code of Ethics for Tourism” (GCET) has been fulfilled in the country”

CORRECTED

Reviewer 3 Report

Dear Authors, 

The introduction must be written again in a standard format of INTRODUCTION. it does not have related paragraphs etc. 

Also, it is not the right place to define: The ten principles of GCET. 

Regarding the literature review, it seems you are teaching something at the beginning to the readers about some fundamental definitions!!!! which is not suggested at all in a paper! 

Why the pilot study and main participants are the visitors while the research questions are asking about the local point of view?!!!!

define the way that you reached the statements in Table.4. 

You must find and define a logical relationship between your research questions and results with GCET! no footprint of that in methodology and data collection and suddenly it appears in all the parts of the conclusion! There is not any integration. 

Author Response

The introduction must be written again in a standard format of INTRODUCTION. it does not have related paragraphs etc. 

CORRECTED

Also, it is not the right place to define: The ten principles of GCET. 

CORRECTED

Regarding the literature review, it seems you are teaching something at the beginning to the readers about some fundamental definitions!!!! which is not suggested at all in a paper! 

CORRECTED

Why the pilot study and main participants are the visitors while the research questions are asking about the local point of view?!!!!

THE MAIN PARTICIPANTS ARE NOT THE TOURISTS. THE QUESTIONNAIRES WERE ADDRESSED TO THE RESIDENTS OF THE CITY OF MUMBAI AS WELL AS TO THE TOURISTS. IT IS CLEARLY MENTIONED IN THE METHODOLOGY, FIRST PARAGRAPH.

define the way that you reached the statements in Table.4. 

Statements are justified on methodology

You must find and define a logical relationship between your research questions and results with GCET! no footprint of that in methodology and data collection and suddenly it appears in all the parts of the conclusion! There is not any integration. 

CORRECTED

ORIGINALITY REPORT

1

NO ALTERATION TO BE MADE. IT IS A CITATION!

2

CORRECTED

3

-

4

These are names and affiliations. Cannot be altered!

5

NO ALTERATION TO BE MADE

6

NO ALTERATION TO BE MADE

7

GCET PRINCIPLES ARE ALREADY WRITTEN, WE CANNOT CHANGE THEM

8

NO ALTERATION TO BE MADE

9

NO ALTERATION TO BE MADE. IT IS A CITATION!

10

NO ALTERATION TO BE MADE. IT IS A CITATION!

11

CORRECTED

12

THESE ARE STATISTICAL RESULTS. CANNOT BE CHANGED

13

NO ALTERATION TO BE MADE. IT IS A CITATION!

14

THESE ARE STATISTICAL RESULTS. CANNOT BE CHANGED

15

THESE ARE STATISTICAL RESULTS. CANNOT BE CHANGED

16

NO ALTERATION TO BE MADE. IT IS A CITATION!

17

NO ALTERATION TO BE MADE. IT IS A CITATION!

18

NO ALTERATION TO BE MADE. IT IS A CITATION!

19

NO ALTERATION TO BE MADE. IT IS A CITATION!

20

NO ALTERATION TO BE MADE

21

NO ALTERATION TO BE MADE. IT IS A CITATION!

22

THESE ARE STATISTICAL PARAMETERS. CANNOT BE CHANGED

23

NO ALTERATION TO BE MADE. IT IS A CITATION!

24

NAMES CANNOT BE CHANGED

25

NAMES CANNOT BE CHANGED

26

THESE ARE STATISTICAL RESULTS. CANNOT BE CHANGED

27

NO ALTERATION TO BE MADE. IT IS A CITATION!

28

THESE ARE STATISTICAL RESULTS. CANNOT BE CHANGED

29

NO ALTERATION TO BE MADE

30

NO ALTERATION TO BE MADE. IT IS A CITATION!

31

THESE ARE STATISTICAL RESULTS. CANNOT BE CHANGED

32

THESE ARE STATISTICAL RESULTS. CANNOT BE CHANGED

33

THESE ARE STATISTICAL RESULTS. CANNOT BE CHANGED

34

NO ALTERATION TO BE MADE

35

THESE ARE STATISTICAL RESULTS. CANNOT BE CHANGED

36

GCET PRINCIPLES ARE ALREADY WRITTEN, WE CANNOT CHANGE THEM

37

THESE ARE STATISTICAL PARAMETERS. CANNOT BE CHANGED

38

NO ALTERATION TO BE MADE. IT IS A CITATION!

39

THESE ARE STATISTICAL RESULTS. CANNOT BE CHANGED

40

NO ALTERATION TO BE MADE. IT IS A CITATION!

41

NO ALTERATION TO BE MADE. IT IS A CITATION!

42

NO ALTERATION TO BE MADE. IT IS A CITATION!

43

NO ALTERATION TO BE MADE. IT IS A CITATION!

44

THESE ARE STATISTICAL RESULTS. CANNOT BE CHANGED

45

NO ALTERATION TO BE MADE. IT IS A CITATION!

46

GCET PRINCIPLES ARE ALREADY WRITTEN, WE CANNOT CHANGE THEM

47

THESE ARE STATISTICAL PARAMETERS. CANNOT BE CHANGED

48

NO ALTERATION TO BE MADE

49

NO ALTERATION TO BE MADE

50

NO ALTERATION TO BE MADE. IT IS A CITATION!

51

GCET PRINCIPLES ARE ALREADY WRITTEN, WE CANNOT CHANGE THEM

52

GCET PRINCIPLES ARE ALREADY WRITTEN, WE CANNOT CHANGE THEM

53

THESE ARE STATISTICAL RESULTS. CANNOT BE CHANGED

54

NO ALTERATION TO BE MADE

55

THESE ARE STATISTICAL RESULTS. CANNOT BE CHANGED

56

NO ALTERATION TO BE MADE. IT IS A CITATION!

57

NO ALTERATION TO BE MADE. IT IS A CITATION!

58

NO ALTERATION TO BE MADE

59

NO ALTERATION TO BE MADE

60

NAMES CANNOT BE CHANGED

61

NO ALTERATION TO BE MADE. IT IS A CITATION!

The introduction must be written again in a standard format of INTRODUCTION. it does not have related paragraphs etc. 

CORRECTED

Also, it is not the right place to define: The ten principles of GCET. 

CORRECTED

Regarding the literature review, it seems you are teaching something at the beginning to the readers about some fundamental definitions!!!! which is not suggested at all in a paper! 

CORRECTED

Why the pilot study and main participants are the visitors while the research questions are asking about the local point of view?!!!!

THE MAIN PARTICIPANTS ARE NOT THE TOURISTS. THE QUESTIONNAIRES WERE ADDRESSED TO THE RESIDENTS OF THE CITY OF MUMBAI AS WELL AS TO THE TOURISTS. IT IS CLEARLY MENTIONED IN THE METHODOLOGY, FIRST PARAGRAPH.

define the way that you reached the statements in Table.4. 

Statements are justified on methodology

You must find and define a logical relationship between your research questions and results with GCET! no footprint of that in methodology and data collection and suddenly it appears in all the parts of the conclusion! There is not any integration. 

CORRECTED

ORIGINALITY REPORT

1

NO ALTERATION TO BE MADE. IT IS A CITATION!

2

CORRECTED

3

-

4

These are names and affiliations. Cannot be altered!

5

NO ALTERATION TO BE MADE

6

NO ALTERATION TO BE MADE

7

GCET PRINCIPLES ARE ALREADY WRITTEN, WE CANNOT CHANGE THEM

8

NO ALTERATION TO BE MADE

9

NO ALTERATION TO BE MADE. IT IS A CITATION!

10

NO ALTERATION TO BE MADE. IT IS A CITATION!

11

CORRECTED

12

THESE ARE STATISTICAL RESULTS. CANNOT BE CHANGED

13

NO ALTERATION TO BE MADE. IT IS A CITATION!

14

THESE ARE STATISTICAL RESULTS. CANNOT BE CHANGED

15

THESE ARE STATISTICAL RESULTS. CANNOT BE CHANGED

16

NO ALTERATION TO BE MADE. IT IS A CITATION!

17

NO ALTERATION TO BE MADE. IT IS A CITATION!

18

NO ALTERATION TO BE MADE. IT IS A CITATION!

19

NO ALTERATION TO BE MADE. IT IS A CITATION!

20

NO ALTERATION TO BE MADE

21

NO ALTERATION TO BE MADE. IT IS A CITATION!

22

THESE ARE STATISTICAL PARAMETERS. CANNOT BE CHANGED

23

NO ALTERATION TO BE MADE. IT IS A CITATION!

24

NAMES CANNOT BE CHANGED

25

NAMES CANNOT BE CHANGED

26

THESE ARE STATISTICAL RESULTS. CANNOT BE CHANGED

27

NO ALTERATION TO BE MADE. IT IS A CITATION!

28

THESE ARE STATISTICAL RESULTS. CANNOT BE CHANGED

29

NO ALTERATION TO BE MADE

30

NO ALTERATION TO BE MADE. IT IS A CITATION!

31

THESE ARE STATISTICAL RESULTS. CANNOT BE CHANGED

32

THESE ARE STATISTICAL RESULTS. CANNOT BE CHANGED

33

THESE ARE STATISTICAL RESULTS. CANNOT BE CHANGED

34

NO ALTERATION TO BE MADE

35

THESE ARE STATISTICAL RESULTS. CANNOT BE CHANGED

36

GCET PRINCIPLES ARE ALREADY WRITTEN, WE CANNOT CHANGE THEM

37

THESE ARE STATISTICAL PARAMETERS. CANNOT BE CHANGED

38

NO ALTERATION TO BE MADE. IT IS A CITATION!

39

THESE ARE STATISTICAL RESULTS. CANNOT BE CHANGED

40

NO ALTERATION TO BE MADE. IT IS A CITATION!

41

NO ALTERATION TO BE MADE. IT IS A CITATION!

42

NO ALTERATION TO BE MADE. IT IS A CITATION!

43

NO ALTERATION TO BE MADE. IT IS A CITATION!

44

THESE ARE STATISTICAL RESULTS. CANNOT BE CHANGED

45

NO ALTERATION TO BE MADE. IT IS A CITATION!

46

GCET PRINCIPLES ARE ALREADY WRITTEN, WE CANNOT CHANGE THEM

47

THESE ARE STATISTICAL PARAMETERS. CANNOT BE CHANGED

48

NO ALTERATION TO BE MADE

49

NO ALTERATION TO BE MADE

50

NO ALTERATION TO BE MADE. IT IS A CITATION!

51

GCET PRINCIPLES ARE ALREADY WRITTEN, WE CANNOT CHANGE THEM

52

GCET PRINCIPLES ARE ALREADY WRITTEN, WE CANNOT CHANGE THEM

53

THESE ARE STATISTICAL RESULTS. CANNOT BE CHANGED

54

NO ALTERATION TO BE MADE

55

THESE ARE STATISTICAL RESULTS. CANNOT BE CHANGED

56

NO ALTERATION TO BE MADE. IT IS A CITATION!

57

NO ALTERATION TO BE MADE. IT IS A CITATION!

58

NO ALTERATION TO BE MADE

59

NO ALTERATION TO BE MADE

60

NAMES CANNOT BE CHANGED

61

NO ALTERATION TO BE MADE. IT IS A CITATION!

,

,

Round 2

Reviewer 1 Report

With the proposed improvements, the article is considered to meet the requirements for publication.

Author Response

We make improvements as requested.

Reviewer 2 Report

Please see the attachment file.

1.       Regarding all of my Previous questions. Please point-by-point provide detailed information about which LINES in your manuscript answer every specific question. So don't just write down the answer "CORRECTED". Because I find it difficult if I have to choose one by one. So until now, I have not been able to judge whether you have answered all my questions well enough or not.

2.       Regarding your answer that “THERE IS NO SUCH REPETITION”, I cannot accept that answer. Here I give you the reason:

a.       In your first manuscript, lines 34, 35, 36, 37, and 38, you wrote :

“Tourism became a cash cow for every country in the world, helping boost consumption, opening of new hotels and related businesses, and providing new jobs to millions of people (Khan, 2014). While the positive effects of tourism are many, it has its negative impacts too, right from irreparable damages to the environment and societies, inflation and ethics.”

b.       Whereas in your first manuscript, lines 100, 101, 102, and 103, you wrote :

“… tourism has become a cash cow for every country in the world, boosting consumption and increasing the number of jobs geometrically, but it causes irreparable damages to the physical environment and societies, influencing prices and giving rise to inflation (Silva, 2020).”

c.       You can see that both are similar. Yes, those are not exactly 100% the same, however the message conveyed is the same, so it's repetitive.

d.       Even I get confused, on lines 34, 35, and 36 as if you are paraphrasing the sentence from Khan (2004). However, on lines 100, 101, 102, and 103 it's as if you are paraphrasing the sentence from Silva (2020). Which one do you actually refer to in the sentence? Did you really read both of them? Therefore, on this occasion, I also ask you to attach the two papers in the next answer, accompanied by colored highlights of the original sentences in each paper. So that I can clearly notice whether it is true that the two papers said the same thing.

3.       Several paragraphs still only consist of one sentence. Avoid a paragraph that consists of only one sentence.

4.       Regarding your answer which said “The survey was built based on scales tested and developed by different authors (Madrigal, 1993; kieti & Magio, 2013; and Mano et al, 2017). As these are multidimensional scales and contain latent variables, different items/statements are used (some of which are similar, its normal) that allow the assessment of the same dimension”

I have to say that I can not buy that answer. Especially for the second sentence. You have to give a valid theoretical basis. Which sources say that it is normal? Which sources state that different items/statements are used because there are multidimensional scales and contain latent variables? Please explain your answer in detail.

5.       Looking at your revised manuscript, I also have several further questions:

a.       In Line 224 of your revised manuscript, you wrote: “The questionnaire had both closed and open-ended questions administered to 202 residents or tourists.”

However in Line 226 and 227, you wrote: “The questionnaire was built on “google forms” and after carrying out a pre-test (Pestana & Gageiro, 2014; Malhotra, 226 2019) with 15 visitors”

Why do only visitors do the pre-test? Why don't residents do a pretest too? In fact, if you look at the instrument used, there are lots of statements made to residents.

b.       If indeed your respondents consist of residents and tourists (visitors), then I cannot see this in Table 2. There is no information about the composition between residents and visitors.

c.       Please explain how you got respondents? Is it random sampling? Is it purposive sampling? Or is it a convenience/haphazard sampling?

If you use random sampling, please explain the assumptions and calculations you used to get the number of respondents. Including whether distributed normally? What is the standard deviation? etc.

If you use purposive sampling, please explain what criteria are used in order to get a respondent. Explain why some were selected and some were not.

If you only use convenience/haphazard sampling, then I think the representativeness of your research is lacking. And in the end, making your research does not guarantee external validity. A lot of scholars agree that convenience sampling should be avoided because it is not representative, one of which is Neuman (2007) in his book entitled "Basics of Social Research Methods: Qualitative and Quantitative Approaches"

Author Response

2ND REVISION

  1. Regarding all of my Previous questions. Please point-by-point provide detailed information about which LINES in your manuscript answer every specific question. So don't just write down the answer "CORRECTED". Because I find it difficult if I have to choose one by one. So until now, I have not been able to judge whether you have answered all my questions well enough or not.

 Point by point justification is given on the table below.

  1. Regarding your answer that “THERE IS NO SUCH REPETITION”, I cannot accept that answer. Here I give you the reason:
  2. In your first manuscript, lines 34, 35, 36, 37, and 38, you wrote :

“Tourism became a cash cow for every country in the world, helping boost consumption, opening of new hotels and related businesses, and providing new jobs to millions of people (Khan, 2014). While the positive effects of tourism are many, it has its negative impacts too, right from irreparable damages to the environment and societies, inflation and ethics.”

  1. Whereas in your first manuscript, lines 100, 101, 102, and 103, you wrote :

“… tourism has become a cash cow for every country in the world, boosting consumption and increasing the number of jobs geometrically, but it causes irreparable damages to the physical environment and societies, influencing prices and giving rise to inflation (Silva, 2020).”

  1. You can see that both are similar. Yes, those are not exactly 100% the same, however the message conveyed is the same, so it's repetitive.
  2. Even I get confused, on lines 34, 35, and 36 as if you are paraphrasing the sentence from Khan (2004). However, on lines 100, 101, 102, and 103 it's as if you are paraphrasing the sentence from Silva (2020). Which one do you actually refer to in the sentence? Did you really read both of them? Therefore, on this occasion, I also ask you to attach the two papers in the next answer, accompanied by colored highlights of the original sentences in each paper. So that I can clearly notice whether it is true that the two papers said the same thing.

Silva (2020)´s citation was removed from text and bibliography. Now only Khan (2014) remains.

  1. Several paragraphs still only consist of one sentence. Avoid a paragraph that consists of only one sentence.

There is no paragraph consisting of one sentence in the entire text. There are some sentences that are one-liners, but combining them with the next or previous sentence, does not change their effect or meaning.

  1. Regarding your answer which said “The survey was built based on scales tested and developed by different authors (Madrigal, 1993; kieti & Magio, 2013; and Mano et al, 2017). As these are multidimensional scales and contain latent variables, different items/statements are used (some of which are similar, its normal) that allow the assessment of the same dimension”

I have to say that I can not buy that answer. Especially for the second sentence. You have to give a valid theoretical basis. Which sources say that it is normal? Which sources state that different items/statements are used because there are multidimensional scales and contain latent variables? Please explain your answer in detail.

Perhaps we were not explicit in the justification we gave. What we meant is that we used scales developed and tested by other authors, adopting the same items/statements that they used, as shown in table 1 (Questionnaire statements on slum tourism), with some items being similar (those are not exactly 100% the same, but are similar).

As these are latent variables (they are not directly observable, the manifestations of these variables are only object of observation), the scale uses different observable variables to infer this dimension (in this case, different statements/indicators are used to assess the latent variable) and which correspond to factors such as opinions, attitudes, perceptions, behaviors and value judgments, which can be observed through questions/statements in a questionnaire.

SOURCES:

Borsboom, D., Mellenbergh, G. J., & van Heerden, J. (2003). The theoretical status of latent variables. Psychological Review, 110(2), 203–219. https://doi.org/10.1037/0033-295X.110.2.203

Vilares, M. & Coelho, P. (2005). Satisfação e Lealdade do Cliente. Lisboa: Escolar Editora

Pestana, M.; Gageiro, J. (2014). Análise de Dados para Ciências Sociais: A Complementaridade do SPSS, 2014, 6 ª Edição. Lisboa: Ed. Sílabo

Marôco, J. (2021). Analise De Equaçoes Estruturais: Fundamentos Teóricos, Software & Aplicações (3.a ed.). Pedro Pinheiro: Reporternumber. ISBN 9789899676367

  1. Looking at your revised manuscript, I also have several further questions:
  2. In Line 224 of your revised manuscript, you wrote: “The questionnaire had both closed and open-ended questions administered to 202 residents or tourists.”

However in Line 226 and 227, you wrote: “The questionnaire was built on “google forms” and after carrying out a pre-test (Pestana & Gageiro, 2014; Malhotra, 226 2019) with 15 visitors”

Why do only visitors do the pre-test? Why don't residents do a pretest too? In fact, if you look at the instrument used, there are lots of statements made to residents.

In fact, as mentioned in the paper, the pre-test was only carried out with the visitors. The aim of this preliminary study was to detect poorly formulated questions, forgetfulness, ambiguities, and any other problems that the questions may raise. We agree that it would also be opportune to include the residents. As the questionnaire has already been administered and the data collected, we placed this question in the limitations of the research. Please see lines 479 to 481.

  1. If indeed your respondents consist of residents and tourists (visitors), then I cannot see this in Table 2. There is no information about the composition between residents and visitors.

Respondents are only residents of the city of Mumbai and a few tourists to the city, not slum tourists or slum dwellers (lines 590 and 591).

The research did not intend to identify differences between these two groups, but, as mentioned in lines 192, 193, one of the objectives was “(iv) identify differences in attitudes towards slum tourism as a function of gender and the experience (or not) of visiting a slum”.

Therefore, the table only includes information about nationality (India or other) and whether or not you visited the slum.

  1. Please explain how you got respondents? Is it random sampling? Is it purposive sampling? Or is it a convenience/haphazard sampling?

If you use random sampling, please explain the assumptions and calculations you used to get the number of respondents. Including whether distributed normally? What is the standard deviation? etc.

If you use purposive sampling, please explain what criteria are used in order to get a respondent. Explain why some were selected and some were not.

If you only use convenience/haphazard sampling, then I think the representativeness of your research is lacking. And in the end, making your research does not guarantee external validity. A lot of scholars agree that convenience sampling should be avoided because it is not representative, one of which is Neuman (2007) in his book entitled "Basics of Social Research Methods: Qualitative and Quantitative Approaches"

This is a non-probabilistic and non-random sampling technique (convenience sampling), as mentioned in line 210, 211 (“…a non-probabilistic convenience sample (Malhotra, 2019)”, so it is not representative of the universe, being, therefore, impossible to generalize the conclusions of the study (lack of external validity). We mentioned that in the limitations of the research. Please see lines 479 to 481.

The questionnaire was shared online by one of the researchers residing in India, through his network of contacts, by a group of individuals who knew the “slum” (residents and visitors).

Comments of authors

Reviewer 2:

Please check your dictionary, is the word “slumdweller” connected or separated? In the Keywords section you write it as “slumdweller”, but in the Literature Review section, you write it as “slum dweller”

CORRECTED. The word is now uniform as slum dweller in the entire text.

In the introduction, you should explain in detail, why choose slum tourism as the subject. Why don't you choose another type of uncommon tourism? For example why you did not choose dark tourism. Explain in detail why slum tourism is very important to research

CORRECTED. See lines 43 to 47

Explain in the Introduction section why you made Dharavi your object of research, and why not other slum areas.

CORRECTED See lines 48 to 50

Your title seems very general. Even though you only explore one tourism area. Therefore it is better to narrow the title and focus on Dharavi. This of course shows that your contribution is very narrow. Therefore, in order to make your contribution big enough, you should cover several areas of the slum in India. Not just one area. Use a valid statistical method to choose which slum area to sample.

CORRECTED. Title changed. Not possible to cover other areas of India. Reasons given on lines 48 to 50

The first three paragraphs of The Beginning sub-section seem completely out of touch with your research. Impressed you are just adding useless paragraphs

CORRECTED. Removed. It was an uploading mistake that copied automatically the instructions of MDPI

Avoid a paragraph that consists of only one sentence. For example on Page 2 Line 85, there is only one sentence in the paragraph. It's a good idea to check and correct all other paragraphs.

CORRECTED No paragraph has only one sentence

Sentences on lines 100, 101, 102, and 103 seem to be just replications and paraphrases of sentences on lines 34, 35, 36, 37, and 38. It's best to avoid repetitive sentences. Very uncomfortable to read

THERE IS NO SUCH REPETITION. Deleted Silva (2020) citation, and now only Khan (2014) remains

You have to explain, where each statement instrument in the questionnaire comes from? If you paraphrase from another source, then you should include the citation in each statement. For example, where does the statement “Slum tourism (ST) is a pleasure” come from? So every statement must be clearly sourced by academic reasons. Because there may be other positive or negative statements that have not been accommodated. For example, regarding negative statements, the statement “Slum Tourism can damage local culture” should also be included

CORRECTED Please see explanation above

Also, some statements seem to mean the same thing. For example, the statement “Resident population obtains economic benefits from ST” and the statement “Tourism can have economic advantages to slums and local entrepreneurs” have the same meaning that is about economical benefits. You must explain why the two things should be separated.

The survey was built based on scales tested and developed by different authors (Madrigal, 1993; kieti & Magio ,2013; and Mano et al, 2017). As these are multidimensional scales and contain latent variables, different items/statements are used (some of which are similar, its normal) that allow the assessment of the same dimension

Perhaps we were not explicit in the justification we gave. What we meant is that we used scales developed and tested by other authors, adopting the same items/statements that they used, as shown in table 1 (Questionnaire statements on slum tourism), with some items being similar (those are not exactly 100% the same, but are similar).

As these are latent variables (they are not directly observable, the manifestations of these variables are only object of observation), the scale uses different observable variables to infer this dimension (in this case, different statements/indicators are used to assess the latent variable) and which correspond to factors such as opinions, attitudes, perceptions, behaviors and value judgments, which can be observed through questions in a questionnaire.

SOURCES:

Borsboom, D., Mellenbergh, G. J., & van Heerden, J. (2003). The theoretical status of latent variables. Psychological Review, 110(2), 203–219. https://doi.org/10.1037/0033-295X.110.2.203

Vilares, M. & Coelho, P. (2005). Satisfação e Lealdade do Cliente. Lisboa: Escolar Editora

Pestana, M.; Gageiro, J. (2014). Análise de Dados para Ciências Sociais: A Complementaridade do SPSS, 2014, 6 ª Edição. Lisboa: Ed. Sílabo

Marôco, J. (2021). Analise De Equaçoes Estruturais: Fundamentos Teóricos, Software & Aplicações (3.a ed.). Pedro Pinheiro: Reporter number. ISBN 9789899676367

Add a Discussion Chapter that provides an in-depth interpretation of your research results. Compare with the results of previous studies. Give reasonable insight. Provide some future direction. Also, explain the advantages of your research results compared to previous studies. The discussion section is the most important part of a scientific paper because it shows the depth and breadth of the researcher in understanding the research topic. At this time, it seems that your paper is very under-contributed because this section does not exist

CORRECTED AND INCLUDED. The discussion section starts in line 326

You have to explain that with the various negative impacts, then how to reduce or eliminate these negative impacts.

CORRECTED. You can see it in lines 469 to 474

From the start, you said you would analyze based on CGET, but in fact, it only appears at the conclusion. And the contents seem just your personal opinion. Each of these GCET points should be discussed with valid and complete data. Should not be written in the Conclusion section, but discussed in the results chapter and discussion chapter.

CORRECTED. You can see it on table 8 discussions and in the discussions section

Page 1 Line 22, “….like slum tourism, dark tourism, sex tourism”. It's better to add “and” before the “sex tourism” phrase.

CORRECTED See lines 22 and 23

In the abstract, please make this sentence more concise. I did not understand the current form:

“This paper analyzes the case of slum tourism to Dharavi, India´s commercial capital and largest city and its benefits and drawbacks, as well as the opinion of the common people and slum-dwellers, while observing if the ten principles of “Global Code of Ethics for Tourism” (GCET) has been fulfilled in the country”

CORRECTED It has been altered in the abstract

You must find and define a logical relationship between your research questions and results with GCET! no footprint of that in methodology and data collection and suddenly it appears in all the parts of the conclusion! There is not any integration. 

CORRECTED You can see it on table 8 discussions and in the discussions section

,

Reviewer 3 Report

Dear authors thanks for your effort to improve and enrich the whole manuscript. 

Author Response

We make improvements as requested.

Round 3

Reviewer 2 Report

I see that many improvements have been made by the authors. A great effort that deserves appreciation.

However, there are still several paragraphs in the discussion section which only consist of one sentence. The author should try to elaborate the paragraph better. Paragraphs should be 3-5 sentences, with the occasional exception. And not just combining several paragraphs into one, but also making sure that every sentence in the paragraph supports one idea.

Most of your manuscript is perfectly fine now, but there are a few areas where one paragraph could be expanded. For example, this paragraph in the discussion section "As evidenced in the literature (Chhabra & Chowdury, 2012; Kieti & Magio, 2013; Frenzel et al., 2015; Tzanelli, 2018; Yagi & Frenzel, 2022; Farmaki & Pappas, 2022), our investigation confirmed that the perceived negative impacts of slum tourism are greater than the expected benefits, with few opportunities for favela residents". I suggest you can expand the paragraph, by comparing one by one your result and each of that literature. There may be minor differences that can be picked up. For example, the negative impact proposed by Chhabra & Chowdury (2012), may be different from the negative impact raised by Kieti & Magio (2013).

Please revised all of the other paragraphs which still consist of only one sentence.

Author Response

Thank you for your attention, sending the proposed corrections and improvements in the article itself. Thank you very much, Manuel Sousa Pereira

Round 4

Reviewer 2 Report

There are still several paragraphs that only consist of one sentence. The author should try to elaborate the paragraph better. Because it is not comfortable to read.

However, I leave it to the Editor whether it is worth accepting or rejecting. Because my comments and suggestions about the number of sentences in the paragraph have been submitted since the first revision. But it turns out that not all paragraphs have been corrected until now.